# Effects of the New COVID-19-Induced Rule on Substitutions and Performance in Italian Elite Soccer

**DOI:** 10.3390/sports11110208

**Published:** 2023-10-26

**Authors:** Bruno Ruscello, Matteo Tozzi, Gennaro Apollaro, Alberto Grossi, Gabriele Morganti, Mario Esposito, Laura Pantanella, Giuseppe Messina, Elvira Padua

**Affiliations:** 1Department of Human Sciences and Promotion of the Quality of Life, San Raffaele Roma Open University, 00166 Rome, Italy; gabriele.morganti@uniroma5.it (G.M.); giuseppe.messina@uniroma5.it (G.M.); elvira.padua@uniroma5.it (E.P.); 2School of Sport and Exercise Sciences, Faculty of Medicine and Surgery, Tor Vergata University, 00133 Rome, Italy; matteo.tozzi17@gmail.com (M.T.); gen.2012.ita@hotmail.com (G.A.); albertogrossi09@gmail.com (A.G.); mario.esposito@uniroma2.it (M.E.); laura.pantanella@gmail.com (L.P.); 3Faculty of Medicine, University of Ostrava, 703 00 Ostrava, Czech Republic; 4PLab Research Institute, 90121 Palermo, Italy

**Keywords:** temporal patterns, tactical adjustments, kinematic components, soccer ecosystem, sprint performance, pandemic era

## Abstract

COVID-19 has resulted in widespread changes, including within the realm of sports. Professional soccer has adapted by allowing more substitutions, leading to tactical adjustments and potential physical benefits. Accordingly, this study analyzed the impact of the new rule in Italian top-level soccer, focusing on substitution patterns and performance differences between the pre-COVID (2017–2018, 2018–2019 seasons) and post COVID (2020–2021, 2021–2022 seasons) eras. As such, publicly available data from 1520 matches (760 matches per era) were recorded. The sample included matches played from 40 Italian top division teams in both the pre- and post-COVID eras. Analyses confirmed substitutions follow a consistent temporal pattern throughout the match in both eras, highlighting a slight difference in second-half management, and showed the new rule is still not used to its full potential, thus raising concerns about teams’ financial strength, as not all managers possess “deep benches” (i.e., a large number of top-level players available to play). Further analyses revealed a statistically significant increment (*p* = 0.002) in the quantity of collectively produced sprints in the post-COVID era compared to the pre-COVID one. The results from this study emphasize the need to carefully address sprint preparation and repeated sprint abilities, also considering factors such as the number of substitutes and their skill level.

## 1. Introduction

The COVID-19 pandemic has caused, and continues to cause, numerous changes in the living conditions of the inhabitants of our planet, affecting physical activities and sports as well [1,2,3,4,5]. Many professional sports have had to adapt their competitions in response to the restrictions imposed during the pandemic period, partially altering the “ecosystem” of the respective sports disciplines. Naturally, any change within any ecosystem requires appropriate adjustments to withstand the impact of the change [6,7,8,9,10,11,12,13,14].

In soccer, for example, the trend of playing an increasing number of matches in the shortest available time has brought about a significant change in the management of substitutions, promoting a genuine “small revolution” in the soccer environment, which is often reluctant to embrace rule changes [15,16,17,18,19,20,21,22,23]. Due to the COVID-19 pandemic, matches and soccer-specific training were suspended for several weeks. After the resumption, matches became congested, leading to an increase in substitutions per team and per game from three to five [24]. Prior to the COVID-19 pandemic, the maximum number of players that could be replaced was three, allowing a maximum replacement of 30% of the outfield players, excluding the goalkeeper. The new rule, which was implemented during the 2019–2020 Italian soccer season, allowed for the substitution of five players, equivalent to a maximum replacement of 50% of the outfield players, still excluding the goalkeeper. In the 2019–2020 season, as stated by Thron et al. [24], Bundesliga teams managed to maintain their physical match performance despite a 9-week break in matches and a 3-week break in group training. On the other hand, in the other leagues, a longer interruption of up to 15 weeks in matches and 8 weeks in group training seemed to result in a decline in physical match performance. In line with this, it is worth noting how recent studies [18], have addressed the need for introducing a new rule that would permanently increase the number of allowable substitutions, not just during difficulties arising from the pandemic. These studies conclude that elite soccer imposes significantly higher physical demands due to its overall set of rules compared to elite futsal, basketball, and handball. Therefore, a permanent increase in soccer substitutions (e.g., unlimited substitutions) could potentially alleviate the physical demands in soccer overall.

As such, the temporary rule permitting five substitutions has now been established as a permanent regulation, starting from the 2022–2023 season onwards. This decision was made by the International Football Association Board (IFAB), the governing body responsible for setting football rules, during its 136th Annual General Meeting held in Doha, Qatar in June 2022. This new rule impacted the management of substitutions, introducing new possibilities and providing greater flexibility in the adopted systems and tactical approaches [14,15,25,26,27]. The increase in the number of substitutions allowed in soccer also has the potential to significantly impact the way coaches manage the training load of their players, as physical outputs in soccer training can be influenced by the interaction of several variables, as has been demonstrated in different small-sided games (SSGs) [28]. Previous studies [17,29] have already noted how the increased number of available players can lead to a distinct management of substitutions under the new rule. In summary, these studies have observed a higher frequency of changes during the interval between the first and second half, as well as a tendency to introduce “fresh forces” around the 60th minute of the game. Considering these premises, it was deemed appropriate to address several research questions regarding the impact of this new rule on Italian soccer in the top division (Serie A). Our focus primarily revolves around the following aspects:What are the temporal patterns of substitutions in top-level Italian soccer before and after the implementation of the new rule?How is the new rule utilized in terms of the maximum number of substitutions allowed?Are there significant differences in certain kinematic components of performance when comparing the periods before and after the implementation of the new rule?

Our hypothesis suggests that the introduction of the new rule has caused specific alterations within the Italian top-level soccer ecosystem, resulting in measurable adaptations that have partially modified previously established reference models from before the COVID-19 era.

The objective of this study is to assess the magnitude of these adaptations by comparing the pre- and post-COVID-19 scenarios. The aim is to establish contemporary reference models for the current state, which can assist in the decision-making process of various stakeholders involved in this context, including coaches, fitness trainers, sports scientists, and others.

## 2. Materials and Methods

### 2.1. Research Design

This study was conducted in an observational manner by comparing publicly available data obtained from the Italian Lega Serie A platform. Ethical review and approval were waived for this study since the investigated data were publicly accessible. As the data were readily available from public sources, there was no requirement to seek authorization from the original sources or obtain approval from any ethics committee. This study was conducted in accordance with the Declaration of Helsinki of 1975, revised in 2013. In our study, we designated the various measures we obtained as the dependent variables, while treating the pre-COVID-19 period (or pre-COVID era) and the post-COVID-19 period (or post-COVID era) as the independent variables.

### 2.2. Sampling

The open-source data for this study were collected from the https://www.legaseriea.it/it/serie-a official site (accessed on 15 July 2022) and encompassed the soccer seasons of 2017–2018, 2018–2019 (pre-COVID era), 2020–2021, and 2021–2022 (post-COVID era). In this latter season, the observations were conducted only until the 19th matchday, as from the 20th matchday, the Italian Lega Serie A changed the measurement parameters that had been previously adopted.

All teams participating in the Italian top division championship (Serie A) were included, resulting in a total sample size of 40 teams in both the pre-COVID and post-COVID eras each.

Considering the dynamic nature of the soccer ecosystem, characterized by ongoing changes within individual teams in terms of personnel mobility (players, coaches, trainers, etc.), we treated the pre-COVID teams from the 2017–2018 and 2018–2019 seasons as single independent samples (referred to as the pre-COVID era sample). Similarly, the teams participating in the 2020–2021 and 2021–2022 seasons were considered single independent samples (referred to as the post-COVID era sample).

For the purpose of this study, only league matches that did not result in red cards (definitive expulsions) were considered. A total of 20 teams per competitive season (80 teams in total) were analyzed, involving approximately 500 players each year. Each season comprised 380 matches, resulting in a total of 1520 matches analyzed (760 matches per era).

### 2.3. Variables Included in this Study

In this study, we considered several variables, which we grouped into three distinct subsections.

For the subsection “Temporal Patterns of Substitutions”, we considered the following variables:The number of substitutions made in the different seasons considered, both pre- and post-COVID (dependent variables).Time blocks of 5 min each (9 in the first half and 10 in the second half, considering stoppage time—independent variable).

For the subsection “Adherence and use of the new rule”, we considered the following variables:Absolute number of substitutions made (n).Normalized number of substitutions as a percentage (%) relative to the maximum possible number of substitutions per game in the different sport seasons considered.

For the subsection “Influence of the number of substitutions on the kinematic components of performances”, we considered the following dependent variables:Total kilometers covered by the team (km).Average speed of the team (km/h).Total kilometers covered by the team in jogging (km).Total kilometers covered by the team in running (km).Total kilometers covered by the team in sprinting (km).

We considered the different sports seasons both before and after the pandemic period with the introduction of the new substitution rule as independent variables.

### 2.4. Data Analysis

Data are presented as mean and standard deviations (M ± SD), along with 95% confidence intervals (95% CIs). Absolute and relative frequencies are also provided when opportune. The normality assumption was assessed using the Kolmogorov–Smirnov test. To verify possible differences in the temporal distribution of substitutions made in the pre-COVID-19 and post-COVID-19 eras, the Chi-Square test was employed using contingency tables. To assess significant differences in some kinematic components of the physical performance between the groups (pre-COVID era vs. post-COVID era), a one-way Analysis of Variance (ANOVA) was performed, considering the recorded measures of performance as dependent variables and the COVID eras as independent ones. Post hoc analyses were performed according to the Bonferroni multiple-comparison correction method. In addition to null hypothesis testing, effect sizes (ω^2^) were reported. Absolute effect sizes of 0.01, 0.06, and 0.14 were classified as small, medium, and large differences, respectively [30]. Corresponding *p*-values were provided for each analysis, and a significance level of *p* < 0.05 was accepted. IBM SPSS 27 for Windows (SPSS Inc., Chicago, IL, USA) was employed to analyze and process the collected data.

## 3. Results

In this section, we present the findings that emerged in response to the research questions posed in the introduction. For the sake of clarity, we report the results in the following subsections.

### 3.1. Temporal Patterns of Substitutions

To analyze and compare the temporal patterns of substitutions utilized (3 vs. 5), we examined two complete sports seasons, pre- and post-COVID-19: 2018–2019 and 2020–2021. The seasons reported in Table 1 and Table 2 were considered for illustrative purposes only, as representatives of the last pre-COVID-19 season and the first post-COVID-19 season, respectively. In this case, our interest was to investigate the immediate effect of the new rule on game management.

We evaluated time blocks of 5 min each (9 blocks of 5 min in the first half and 10 blocks of 5 min in the second half, including stoppage time).

A total of 2209 and 3276 substitutions were observed in season 2018–2019 and 2020–2021, respectively, and allocated within their respective time blocks.

The implementation of the new substitution rule resulted in a significant increase in the observed absolute values of substitutions (2018–2019, n = 2209; 2020–2021, n = 3276, with an absolute difference of 1067, corresponding to a 48.3% increase). The observed absolute frequencies within the blocks were then evaluated, normalizing them to relative percentage frequencies based on the total observed substitution values in the respective sports seasons (see Table 1 and Table 2).

To evaluate possible significant differences in the distribution of substitutions made in the two seasons considered, a Chi-Squared test, normalized to their percentage values, was used through a contingency table.

The results obtained confirm that, from a percentage perspective, the relative behaviors align in the two seasons, confirming no significant differences, considering the two halves:

First half 2018–2019 vs. 2020–2021 (time blocks 1–5 min up to 41–45 min): Chi-Squared Test = 3.733; df = 8; *p* = 0.880.

Second half 2018–2019 vs. 2020–2021 (time blocks: 46–50 min up to 91–95 min): Chi-Square Test = 0.894; df = 9; *p* = 1.000; Cramer’s V = 0.068.

### 3.2. Adherence and Use of the New Rule

As stated in the research question, this study aimed to address a topic of practical interest: the extent of utilization of the new rule in terms of the use of potential substitutions, following the transition from 3 to 5 substitutions. In this case, four sports seasons were analyzed: two pre-COVID-19 seasons (2017–2018 and 2018–2019) and two post-COVID-19 seasons (2020–2021 and 2021–2022). To obtain information about this point, we have reported the absolute number of substitutions made (n) and normalized them as a percentage relative to the maximum possible theoretical number. The descriptive statistical findings are provided in Table 3.

### 3.3. Influence of the Number of Substitutions on the Kinematic Components of Performances

We have analyzed the various measurements obtained on some kinematic parameters, reported in the databases used in this study, which compare four different sport seasons: 2017–2018, 2018–2019, 2019–2020 and 2021–2022.

1. Total kilometers covered by the team (km): This refers to the overall distance covered by the team during the specified sport seasons. It represents the sum of distances covered during jogging, running, and sprinting.

2. Average speed by the team (km/h): This refers to the average speed at which the team collectively covered the measured distance during the time of the match.

3. Total kilometers covered in jogging by the team (km): This indicates the distance covered by the team while jogging. Jogging typically involves running at a moderate pace ranging from 0.1 m per second (m/s) up to approximately 1.66 m/s.

4. Total kilometers covered in running by the team (km): This represents the distance covered by the team while running. Running generally involves a faster pace than jogging, ranging from 1.67 m/s up to 5.55 m/s.

5. Total kilometers covered in sprinting by the team (km): This refers to the distance covered by the team during sprinting, which involves high-speed running. The sprinting speed range mentioned starts from 5.56 m/s and extends to even higher speeds. Table 4 provides relevant descriptive statistics for the kinematic components analyzed throughout the four-year period under investigation.

In order to verify possible differences among the investigated kinematic components in relation to the four analyzed sport seasons, a one-way ANOVA was conducted. The obtained measurements were treated as the dependent variables, while the considered sport seasons were treated as the independent variables.

Statistically significant differences were not observed, based on the given *p*-value, in 4 out of 5 variables considered (total distance, average speed, jogging, and running). However, it is worth noting the *p*-values obtained, which can be considered borderline statistically significant and hold some degree of interest. Particularly, the 95% CI upper values of omega-squared exhibit some practical relevance:
Total distance: F _3,76_= 2.460; *p* = 0.069; ω^2^ = 0.052 [95% CI = −0.039–0.163]Average speed: F _3,76_= 1.627; *p* = 0.129; ω^2^ = 0.023 [95% CI = −0.039–0.122]Jogging: F _3,76_= 2.533; *p* = 0.063; ω^2^ = 0.054 [95% CI = −0.039–0.167]Running: F _3,76_= 2.484; *p* = 0.067; ω^2^ = 0.053 [95% CI = −0.039–0.165].


A different approach is warranted when considering the data related to distances covered in sprinting, as it holds significance from a technical–tactical perspective, representing the qualitative aspect of athletes’ performance. Notably, statistically significant differences have been observed in this kinematic variable, with a large effect size:
5.Sprinting: F _3,76_= 5.355, *p* = 0.002; ω^2^ = 0.140 [95% CI = −0.010–0.270] (see Figure 1).


Post hoc tests using the Bonferroni correction confirmed significant differences in the “sprinting” variable when comparing the 2017–2018 and 2018–2019 seasons with the 2021–2022 season (*p* = 0.011 and *p* = 0.010, respectively).

## 4. Discussion

To the best of our knowledge, this is the first study to analyze the impact of the new substitution rule introduced in Italian elite soccer (Serie A) after the COVID-19 pandemic. This topic has been recognized throughout the global soccer ecosystem, but to date, little is known about this subject, with only a few targeted studies available [15,16,17,18,19,25,26,31,32,33]. The hypotheses we formulated in response to our research questions guided our investigation of three main topics, which we have summarized in the following subsections, structured similarly to the Results section.

### 4.1. Temporal Patterns of Substitution

Our initial hypothesis was to find a different management of substitutions throughout the match, considering the significant potential increase possible with the introduction of the new rule. However, this data cannot be confirmed in light of the statistical analysis conducted (Chi-Squared test; *p* > 0.05). The management of substitutions appears to overlap in the two eras considered—pre- and post-COVID-19—leaving only room for reflection on what is evident, where changes observable in the number and percentage of substitutions made by different management at the beginning of the second half (time block 46–50′) can be considered. Indeed, it is worth highlighting the difference at the start of the second half (time block 46–50 min): 103 in the pre-COVID-19 period (equivalent to 6.02% of the total) and 303 in the post-COVID-19 period (equivalent to 9.25% of the total). This occurrence raises some questions, as it significantly increases the difference, already observed in other studies even before the COVID-19 era, between the two halves of the game [34,35,36]. The management of substitutions observed in this study confirms that the technical–tactical and physical conduct of the game requires well-targeted strategies to mitigate or even overcome collective fatigue phenomena, particularly in the second half of the match. Another relevant aspect is the density of substitutions also observed in our study, where in both the pre-COVID-19 era and the post-COVID-19 era, the time windows most commonly used are those corresponding to the final 30 min of the game [21,22,32].

### 4.2. Adherence and Use of the New Rule

This aspect proves to be of certain importance, as it raises the issue of the wealth disparity between teams. Indeed, the ability to use more players during the management of a soccer match proves to be of significant importance, provided that the quality of the incoming substitutes is at least similar to that of the outgoing players. In coaching terms, this aspect is referred to as having a “deep bench” (i.e., a team with many talented substitutes who can contribute at any time during the match, within the rules of the game). The topic of performance quality is closely related to this concept of a “deep bench”, which naturally implies the financial strength of a club that can afford top-level players who are not in the starting lineup. Not all teams can afford it, even in the Italian Serie A. Strategically planning player substitutions during a soccer match to maintain collective performance quality is important, but it could have a detrimental effect if the substitute’s technical ability is significantly lower than the player they are replacing.

Perhaps for this reason, the percentage utilization rates compared to the maximum theoretical numbers that we found in the pre-COVID-19 seasons (97.3%) are higher than those observed in the post-COVID-19 seasons (87.25%), where five theoretical substitutions are available. This new playing space could be an excellent opportunity to promote youth talent, as more and more coaches are interested in showcasing young athletes from youth academies [37,38]. Introducing a new regulatory intervention, such as a mandatory rule requiring teams to use young players, could potentially be a desirable development of the five-substitution rule.

### 4.3. Influence of the Number of Substitutions on the Kinematic Components of Performances

We investigated kinematic components in open-source databases to determine whether the increased availability of fresher players could impact overall team performance.

Extensive research has investigated soccer performance from a physical standpoint [36,39,40,41,42,43,44,45,46]. This research has shown that the sprint ability (SA) and the repeated sprint ability (RSA) are two of the most important qualitative aspects of performance from a physical perspective [47,48,49,50,51,52,53,54].

The results observed have confirmed that there is a certain diversity in evaluated physical performances, with borderline statistical values (*p* ≈ 0.07), for jogging and running abilities, as well as for total distance covered. This is in the context of the increased number of possible substitutions in the post-COVID-19 era. The findings of this study motivate future researchers to delve into these interesting emerging trends. However, we find the values related to the amount of sprinting observed to be of particular interest, with evident practical implications. These values reveal, with robust statistical significance (*p* = 0.002) and a large effect size (ω^2^ = 0.140), that this characteristic, considered qualitatively important for soccer performance, may have benefited from the implementation of the new rule. This, of course, raises the question of how to develop sprint ability (SA) and repeated sprint ability (RSA) in line with the growth trend observed, which poses new challenges for specific training.

## 5. Practical Applications

The implementation of the new substitution rule in the post-COVID-19 era has presented new demands in the overall management of the soccer ecosystem. This raises the issue of adaptation in both match management and weekly training. In an applied context, we consider it appropriate to propose certain practices that are currently being employed by some professional teams to mitigate the fixture congestion that we have observed in recent years. Workload management during the training week and in periods of fixture congestion has been restructured to involve an integrated evaluation of various staff areas, such as tactics, technique, fitness, medical, match analysis, and performance analysis. Training sessions must be designed based on different training domains, the desired stimulus, and the training cycle. This should also consider the training workload trends and individual-specific locomotor demands.

The evolution of the sports calendar has led to shorter weekly training sessions due to the increase in the number of commitments (friendly matches, league games, cup matches). The need to optimize physical, technical, tactical, and psychological components in a single training session is essential. Small-sided games are an excellent strategic tool to meet these demands. If the coaching staff’s objective is to achieve a reduction in training load (e.g., for recovery, tactical objectives, learning game principles, or introducing a new exercise), the continuous method is advisable. If the goal is to increase external load (running distance, speed, number of sprints, accelerations, and decelerations) or internal load (HR, sRPE, etc.), then the fractional method is the recommended training method. A 5-vs.-5 Small Sided Game (SSG) in comparison to other formats (e.g., 7-vs.-7, 6-vs.-6, 3-vs.-3, etc.) can be effectively used to modulate the workload [28]. This depends on the manipulation of various related variables, such as field dimensions, number of players, duration, repetitions, recovery, and rules.

## 6. Conclusions

In conclusion, this study explored the impact of the new substitution rule introduced after the COVID-19 pandemic in Italian elite soccer (Serie A). Our findings shed light on several key aspects of the management of substitutions, adherence to the rule, and its influence on kinematic components of performance.

Firstly, our analysis revealed that the temporal patterns of substitutions did not significantly differ between the pre- and post-COVID-19 eras. However, a certain difference has been highlighted in the management of substitutions made at the beginning of the second half, with a significant increase in their number in the post-COVID-19 era. Furthermore, we highlighted the potential significance of the wealth disparity between teams when considering the use of substitutes. The concept of a “deep bench” emerged, which indicates the financial strength of clubs that can afford to have high-quality players who do not feature in the starting lineup. It is important to note that not all teams can afford such a “luxury”, even in the highly competitive Italian Serie A. This study also examined the influence of the increased number of substitutions on some kinematic components of performance. The statistical analysis indicated no significant differences in jogging, running abilities, or total distance covered. However, we observed a notable increase in sprinting performance. This suggests that the implementation of the new substitution rule may have benefited this crucial aspect of soccer performance. These findings motivate further research in the development of sprint ability (SA) and repeated sprint ability (RSA) training.

Overall, this study contributes to the existing body of knowledge by providing insights into the effects of the new substitution rule in Italian elite soccer. It highlights the need for well-planned substitution strategies, the consideration of wealth disparities between teams, and the potential benefits for sprinting performance. Future studies should continue to explore these areas and further refine our understanding of implications and potential developments in the utilization of substitutes in the game of soccer.

## Figures and Tables

**Figure 1 sports-11-00208-f001:**
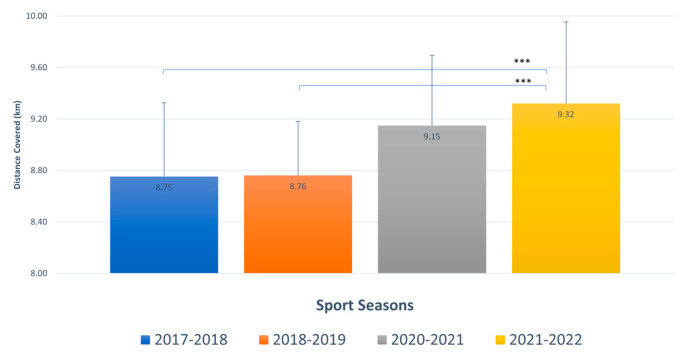
Sprinting: distance covered per team (km) in the different sport seasons. (Mean ± SD). *** = highly significant.

**Table 1 sports-11-00208-t001:** Observed substitutions (absolute frequencies and relative to the maximum possible number of substitutions per game) for each time block considered (data recorded in the first half).

Time Blocks	2018–2019 (n)	2020–2021 (n)	2018–2019 (%)	2020–2021 (%)
01–05 min	2	1	0.09	0.03
06–10 min	2	4	0.09	0.12
11–15 min	13	7	0.59	0.21
16–20 min	8	16	0.36	0.49
21–25 min	16	7	0.72	0.21
26–30 min	15	6	0.68	0.18
31–35 min	9	23	0.41	0.70
36–40 min	12	23	0.54	0.70
41–45 min	14	18	0.63	0.55
**Sum**	91	105	4.12	3.21
**Mean**	10.11	11.66	0.46	0.36
**Standard Deviation**	5.28	8.40	0.24	0.26

n = absolute frequency of substitutions. % = relative frequency of substitutions.

**Table 2 sports-11-00208-t002:** Observed substitutions (absolute values and relative to the maximum possible number of substitutions per game) for each time block considered (data recorded in the second half and stoppage time).

Time Blocks	2018–2019 (n)	2020–2021 (n)	2018–2019 (%)	2020–2021 (%)
46–50 min	133	303	6.02	9.25
51–55 min	76	92	3.44	2.81
56–60 min	180	288	8.15	8.79
61–65 min	241	374	10.91	11.42
66–70 min	277	373	12.54	11.39
71–75 min	315	439	14.26	13.40
76–80 min	337	484	15.26	14.77
81–85 min	319	445	14.44	13.58
86–90 min	195	313	8.83	9.55
91–95 min	45	60	2.04	1.83
**Sum**	2118	3171	95.88	96.79
**Mean**	211.8	371.1	9.59	9.68
**Standard Deviation**	103.64	142.70	4.69	4.36

n = absolute frequency of substitutions. % = relative frequency of substitutions.

**Table 3 sports-11-00208-t003:** Utilization rate of the substitution rule under two different conditions: pre-COVID-19 and post-COVID-19.

Season	Substitutions (n)	Substitutions (% of Maximum Possible Theoretical Number)
2017–2018 ^a^	2229	97.8
2018–2019 ^a^	2209	96.9
2020–2021 ^b^	3276	86.2
2021–2022 ^b^	3357	88.3

^a^ in the 2017–2018 and 2018–2019 seasons, the maximum number of substitutions allowed per match was three players. ^b^ in the 2020–2021 and 2021–2022 seasons, the maximum number of substitutions allowed per match was five players.

**Table 4 sports-11-00208-t004:** Analyzed kinematic components: descriptive statistics.

Variable	Season	Mean	Std. Deviation	Std. Error	95% Confidence Interval for Mean		Minimum	Maximum
					Lower Bound	Upper Bound		
Total distance (km)	2017–2018	109.742	1.727	0.386	108.934	110.550	105.900	112.391
	2018–2019	108.404	1.873	0.419	107.527	109.280	105.723	112.679
	2020–2021	108.950	2.499	0.559	107.780	110.120	102.825	113.257
	2021–2022	107.979	2.465	0.551	106.826	109.133	105.012	114.184
Average Speed (km/h)	2017–2018	6.784	0.117	0.026	6.729	6.838	6.555	6.957
	2018–2019	6.683	0.132	0.030	6.621	6.745	6.471	7.000
	2020–2021	6.727	0.177	0.039	6.644	6.809	6.219	7.019
	2021–2022	6.709	0.167	0.037	6.630	6.787	6.486	7.094
Jogging (km)	2017–2018	27.738	0.794	0.177	27.366	28.109	26.157	28.851
	2018–2019	27.897	0.860	0.192	27.495	28.300	26.450	29.395
	2020–2021	27.204	0.782	0.175	26.838	27.570	25.896	28.404
	2021–2022	27.359	1.142	0.255	26.825	27.894	25.018	28.669
Running (km)	2017–2018	73.251	2.090	0.467	72.273	74.230	68.862	76.221
	2018–2019	71.745	2.354	0.526	70.644	72.847	67.518	77.321
	2020–2021	72.598	2.594	0.580	71.384	73.812	67.129	77.316
	2021–2022	71.300	2.795	0.625	69.992	72.609	67.677	78.425
Sprinting (km)	2017–2018	8.753	0.573	0.128	8.484	9.021	7.958	10.271
	2018–2019	8.761	0.421	0.094	8.564	8.958	8.078	9.613
	2020–2021	9.148	0.546	0.122	8.893	9.404	8.094	10.045
	2021–2022	9.320	0.635	0.142	9.023	9.617	8.126	10.741

Jogging = up to 1.66 m per second (m/s). Running = from 1.67 up to 5.55 m/s. Sprinting = from 5.56 m/s up to higher speeds.

## Data Availability

Raw Data supporting reported results can be found at https://www.legaseriea.it/it/serie-a (accessed on 15 July 2022). Processed data available on request.

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
