# Peer review of "Effects of the New COVID-19-Induced Rule on Substitutions and Performance in Italian Elite Soccer"

_sports, 2023, doi:10.3390/sports11110208_

Round 1

Reviewer 1 Report

Comments and Suggestions for Authors

Examining the Effects of the Covid-19-Induced New Rule on Substitutions and Performance in Italian Elite Soccer: A Comprehensive Analysis

Attempt to condense your title for greater conciseness. You could remove "Examining the" at the very least.

Furthermore, throughout the paper, there is a significant amount of extraneous wording that could be streamlined. Consider having a native scientist review the English editing for accuracy

L26 - be more specific about "deep bench"?

L26-27 - "The results confirm a significant difference (p=0.002) in the quantity of collectively produced sprints before and after the pandemic era."

Encourage the authors to enhance specificity: Could you provide greater detail by indicating whether the number of sprints increased following the change to allow 5 substitutions, as compared to the previous limit of 3 substitutions?

L27-30 - "The impact of the new rule also raises concerns about the quality and quantity of players to be used as substitutes throughout the match, emphasizing the need to address sprint preparation and repeated sprint ability…

Please improve clarity. I am not sure what you mean by "raises concerns about the quality"??

To improve clarity for the reader, it would be beneficial to introduce the variables in the methods section before they are presented in the results section. This way, when readers encounter these variables in the results, they will already have a clear understanding of what they represent due to their prior mention in the methods. This approach will help to establish a smoother and more comprehensible flow of information for the reader.

Consider focusing on presenting results only in the results section, and reserving discussions for the designated section. This will enhance the clarity and organization of your paper.

Revise: Please review the symbol ' used to represent minutes in the tables. Utilize "min" in the title instead and omit the use of '.

Please collectively streamline and sharpen both the discussion and conclusion sections for a more concise and objective presentation of your findings.

Comments on the Quality of English Language

NA

Author Response

For research article

Response to Reviewer X Comments

1. Summary

2. Questions for General Evaluation

Reviewer’s Evaluation

Response and Revisions

Does the introduction provide sufficient background and include all relevant references?

Can be improved

Dear Reviewer, we have modified the section as per your valuable suggestion, including a new paragraph and another citation.

Are all the cited references relevant to the research?

Yes

Thank you for the positive comment

Is the research design appropriate?

Yes

Thank you for the positive comment

Are the methods adequately described?

Can be improved

Dear Reviewer, we have modified the section based on your valuable suggestion. We hope this has improved our work accordingly.

Are the results clearly presented?

Can be improved

Dear Reviewer, we have modified the section based on your valuable suggestion. We hope this has improved our work accordingly.

Are the conclusions supported by the results?

Can be improved.

Dear Reviewer, we have modified the section based on your valuable suggestion. We hope this has improved our work accordingly.

3. Point-by-point response to Comments and Suggestions for Authors

Comments 1: Examining the Effects of the Covid-19-Induced New Rule on Substitutions and Performance in Italian Elite Soccer: A Comprehensive Analysis

Attempt to condense your title for greater conciseness. You could remove "Examining the" at the very least.

Response 1: Dear Reviewer, we have welcomed your valuable suggestion in this regard and have decided to rename the title as follows: “Effects of the Covid-19-Induced New Rule on Substitutions and Performance in Italian Elite Soccer”.

Comments 2: Furthermore, throughout the paper, there is a significant amount of extraneous wording that could be streamlined. Consider having a native scientist review the English editing for accuracy.

Response 2: Dear reviewer, thank you for the feedback. We have undergone a thorough revision of the entire text with the assistance of a native speaker. We hope to have thus overcome this issue.

Comments 3: L26 - be more specific about "deep bench"?

Response 3: Dear reviewer, many thanks for your suggestion we revised the abstract accordingly. In detail, we specified the meaning of the expression “deep bench”, adding to the text the following line: i.e., large number of top-level players available to play. [lines 26-30]

Comments 4: L26-27 - "The results confirm a significant difference (p=0.002) in the quantity of collectively produced sprints before and after the pandemic era."

Encourage the authors to enhance specificity: Could you provide greater detail by indicating whether the number of sprints increased following the change to allow 5 substitutions, as compared to the previous limit of 3 substitutions?

Response 4: Dear Reviewer, many thanks for your suggestion we revised the abstract accordingly.

“Further analyses revealed a statistically significant increment (p=0.002) in the quantity of collectively produced sprints in post-Covid era, compared to the pre-Covid one.” [lines 30-31]

Comments 5: L27-30 - "The impact of the new rule also raises concerns about the quality and quantity of players to be used as substitutes throughout the match, emphasizing the need to address sprint preparation and repeated sprint ability…

Please improve clarity. I am not sure what you mean by "raises concerns about the quality"??

Response 5: Dear reviewer many thanks for your suggestion we revised the abstract accordingly, improving its clarity. Indeed, we completely revised the last part of the abstract and removed the expression “quality of the player”: “Results from this study emphasize the need to address sprint preparation and repeated sprint ability carefully, also considering factors such as number of substitutes and their skill level.” [lines 35-37]

Comments 6: To improve clarity for the reader, it would be beneficial to introduce the variables in the methods section before they are presented in the results section. This way, when readers encounter these variables in the results, they will already have a clear understanding of what they represent due to their prior mention in the methods. This approach will help to establish a smoother and more comprehensible flow of information for the reader.

Response 6: Dear Reviewer, thank you for the suggestion. We have included a subsection titled "Variables Included in the Study" in the Methods section (lines 148-170).

Comments 7: Consider focusing on presenting results only in the results section, and reserving discussions for the designated section. This will enhance the clarity and organization of your paper.

Response 7: Dear Reviewer, thank you for your suggestion. We have revised the Results section in accordance with your guidance, including the relocation of a comment on the data distribution in substitutions from the Results section to the Discussion section.".

Comments 8: Revise: Please review the symbol ' used to represent minutes in the tables. Utilize "min" in the title instead and omit the use of '.

Response 8: Dear Reviewer, thank you for your feedback. We have made textual amendments in the tables (1 and 2) in accordance with your suggestion.

Comments 9: Please collectively streamline and sharpen both the discussion and conclusion sections for a more concise and objective presentation of your findings.

Response 9: Dear Reviewer, thank you for your suggestion. We have revised both the sections you indicated (Discussion and Conclusion), aiming to condense the concepts. However, we also had to take into account the guidance provided by the other reviewer regarding the conclusions of our work: 'Finally, I would like to see a specific point called 'practical applications' inserted, which should contain practical recommendations resulting from the work for the coach. This is indeed a focal point of the work! We trust that this inclusion does not unduly encumber our study.

Comments 10: Comments on the Quality of English Language NA

Response 9: Dear reviewer, we have submitted our paper to a native language expert. We believe that we have achieved a level of communication suitable for the high standards of this journal.

Submission Date 08 August 2023

Date of this review 06 Sep 2023 04:15:31

Reviewer 2 Report

Comments and Suggestions for Authors

Dear authors, first of all thank you for your submission to sports. The authors address a topic of high relevance and that will certainly continue to be studied by other authors, since until now little is known about the real impact that the increase in the number of substitutions effectively has on the management of the game training load of the players. players. Overall the authors did a good job and deserve credit for it, yet there are some recommendations that may help improve the quality of the work. Below you can find my comments in detail.

1) I would like to see a paragraph inserted in the introduction that emphasizes the potential impact that the number of substitutions can have on the management of the training load/playload of the players by the coach throughout the working week and on the management carried out in fixture periods congestion.

I suggest the introduction of the following references in this topic:

- Branquinho, L., Ferraz, R., & Marques, M. C. (2021). 5-a-Side Game as a Tool for the Coach in Soccer Training. Strength and Conditioning Journal, 43(5), 96-108.

2) At the end of the Introduction it describes the dependent and independent variables and in my opinion this information makes more sense in the methods section.

3) Why are data from all periods analyzed not presented in tables 1 and 2?

In figure 1 it is not clear which seasons were used to make the comparison before and after Covid.

These issues related to figures and tables need to be fully clarified.

It is confusing to understand what information is used and why, and why data for all seasons is not presented.

4) Finally, I would like to see a specific point called "practical applications" inserted, which should contain practical recommendations resulting from the work for the coach.

This is indeed a focal point of the work!

Overall good job.

Comments on the Quality of English Language

English at a good level and only minor adjustments needed

Author Response

For research article

Response to Reviewer X Comments

1. Summary

2. Questions for General Evaluation

Reviewer’s Evaluation

Response and Revisions

Does the introduction provide sufficient background and include all relevant references?

Can be improved

Dear Reviewer, we have modified the section as per your valuable suggestion, including a new paragraph and another citation.

Are all the cited references relevant to the research?

Yes

Thank you for the positive comment

Is the research design appropriate?

Yes

Thank you for the positive comment

Are the methods adequately described?

Can be improved

Dear Reviewer, we have modified the section based on your valuable suggestion. We hope this has improved our work accordingly.

Are the results clearly presented?

Can be improved

Dear Reviewer, we have modified the section based on your valuable suggestion. We hope this has improved our work accordingly.

Are the conclusions supported by the results?

Must be improved.

Dear Reviewer, we have modified the section based on your valuable suggestion. We hope this has improved our work accordingly.

3. Point-by-point response to Comments and Suggestions for Authors

General Comments: Dear authors, first of all thank you for your submission to sports. The authors address a topic of high relevance and that will certainly continue to be studied by other authors, since until now little is known about the real impact that the increase in the number of substitutions effectively has on the management of the game training load of the players. players. Overall, the authors did a good job and deserve credit for it, yet there are some recommendations that may help improve the quality of the work. Below you can find my comments in detail.

Response: Dear Reviewer, thank you for your positive feedback. We greatly appreciate it, and it motivates us to give our best in this revision process! We are confident that we can address your suggestions correctly, and with your assistance, we believe we can enhance the quality of our work. Recognizing the essential yet often underappreciated role of the review process, we are well aware of its significance. It contributes not only to our personal growth but also to the advancement of our scientific community. Thank you sincerely.

Comments 1) I would like to see a paragraph inserted in the introduction that emphasizes the potential impact that the number of substitutions can have on the management of the training load/playload of the players by the coach throughout the working week and on the management carried out in fixture periods congestion.

I suggest the introduction of the following references in this topic:

 Branquinho, L., Ferraz, R., & Marques, M. C. (2021). 5-a-Side Game as a Tool for the Coach in Soccer Training. Strength and Conditioning Journal, 43(5), 96-108.

Response 1: Dear Reviewer, we have added a brief paragraph on the topic you mentioned, including the citation of the excellent work you suggested. [lines 85-88 and lines 465-471]

Comments 2) At the end of the Introduction it describes the dependent and independent variables and in my opinion this information makes more sense in the methods section.

Response 2: Dear Reviewer, thank you for your suggestion. We have moved the paragraph you indicated to the Methods section, specifically under the Research Design section. [lines 122-125]

Comments 3) Why are data from all periods analyzed not presented in tables 1 and 2?

In figure 1 it is not clear which seasons were used to make the comparison before and after Covid.

These issues related to figures and tables need to be fully clarified.

It is confusing to understand what information is used and why, and why data for all seasons is not presented.

Response 3: Why are data from all periods analyzed not presented in tables 1 and 2?”. Dear Reviewer, the seasons reported in Tables 1 and 2 were considered for illustrative purposes only, as representatives of the last pre-COVID-19 season and the first post-COVID-19 season, respectively. In this case, our interest was to investigate the immediate effect of the new rule on game management.[lines 197-201]

“In figure 1 it is not clear which seasons were used to make the comparison before and after Covid. These issues related to figures and tables need to be fully clarified.”

Dear Reviewer, we amended figure 1 as per your indication.

Comments 4) Finally, I would like to see a specific point called "practical applications" inserted, which should contain practical recommendations resulting from the work for the coach. This is indeed a focal point of the work!

Response 4: Dear Reviewer, thank you for this suggestion. We completely agree that it is essential to strive to bring scientific research in the field of sports into practical application. For this reason, as per your request, we have added a specific section titled 'practical applications.' We hope it meets your approval. [lines 394-419]

Comments 5) Overall good job.

Response 5: Thank you once again for your positive and flattering assessment of our work. Your feedback is greatly appreciated, and it encourages us to continue with even greater dedication on our research journey.

We are grateful for your advice, and we are confident that, thanks to it, we have been able to improve our work. We will continue to strive to produce high-quality research that contributes to the field of soccer science.

Thank you again for your time and consideration.

Comments 6) on the Quality of English Language

English at a good level and only minor adjustments needed.

Response 6: Dear Reviewer, thank you for the positive feedback. We have further submitted the paper to a native language expert to ensure the highest quality and readability of our work.

Submission Date 08 August 2023

Date of this review 28 Aug 2023 18:56:59

Round 2

Reviewer 1 Report

Comments and Suggestions for Authors

No further comments. 

Author Response

"Dear Reviewer, thank you very much once again for your work. It has allowed us to grow in our understanding of the issue addressed in this research. We will treasure your suggestions in our future research as well."

Reviewer 2 Report

Comments and Suggestions for Authors

The authors did a good job in reviewing all the issues raised by the reviewer, in this sense I have no further reservations regarding the manuscript and recommend its acceptance

Author Response

Dear Reviewer, thank you very much once again for your work. It has allowed us to grow in our understanding of the issue addressed in this research. We will treasure your suggestions in our future research as well.